# Paracrine control of α-cell glucagon exocytosis is compromised in human type-2 diabetes

Muhmmad Omar-Hmeadi [1], Per-Eric Lund[1], Nikhil R. Gandasi [1], Anders Tengholm [1] & Sebastian Barg [1✉]

Glucagon is released from pancreatic α-cells to activate pathways that raise blood glucose. Its secretion is regulated by α-cell-intrinsic glucose sensing and paracrine control through insulin and somatostatin. To understand the inadequately high glucagon levels that contribute to hyperglycemia in type-2 diabetes (T2D), we analyzed granule behavior, exocytosis and membrane excitability in α-cells of 68 non-diabetic and 21 T2D human donors. We report that exocytosis is moderately reduced in α-cells of T2D donors, without changes in voltage-dependent ion currents or granule trafficking. Dispersed α-cells have a non-physiological V-shaped dose response to glucose, with maximal exocytosis at hyperglycemia. Within intact islets, hyperglycemia instead inhibits α-cell exocytosis, but not in T2D or when paracrine inhibition by insulin or somatostatin is blocked. Surface expression of somatostatin-receptor-2 is reduced in T2D, suggesting a mechanism for the observed somatostatin resistance. Thus, elevated glucagon in human T2D may reflect α-cell insensitivity to paracrine inhibition at hyperglycemia.

[1] Medical Cell Biology, Uppsala University, Box 571, BMC, 751 23 Uppsala, Sweden. ✉email: sebastian.barg@mcb.uu.se

Glucagon is released from pancreatic α-cells and counteracts the glucose-lowering actions of insulin by stimulating gluconeogenesis and hepatic glucose output. Initially thought of only as part of the body's defense against hypoglycemia[1], it is now clear that inadequate glucagon levels also contribute to diabetic hyperglycemia and present a challenge for diabetes management[2,3]. Glucagon secretion is triggered by low blood glucose and suppressed at physiological glucose levels, and both α-cell intrinsic and paracrine mechanisms have been cited to explain these effects. In the intrinsic models, glucose metabolism and the generation of ATP play a central role[4–6], either through subtle, $K_{ATP}$ channel-dependent depolarization of the resting membrane potential and subsequent inactivation of $Na^+$-channels[7–10], or as consequence of glucose-induced activation of the sarco/endoplasmic reticulum $Ca^{2+}$-ATPase that leads to closure of store-operated channels and hyperpolarization[11,12]. In addition, intrinsic glucose-dependent cAMP signaling may play a role[13]. However, none of these models fully explain the glucose concentration dependence of glucagon secretion, in particular in the hyperglycemic range.

Glucagon secretion is also under paracrine control from neighboring β- and δ-cells, and the inhibitory effects of somatostatin[14–18], insulin[19–23], and GABA[24,25] have long been recognized. Paracrine inhibition is likely to play a role at elevated glucose levels, when β- and δ-cells are active. Indeed, glucagon is secreted in pulses that are anti-synchronous to pulses of insulin and somatostatin[26,27]. This relationship is important for the postprandial suppression of glucagon secretion and is lost in type-2 diabetes and pre-diabetes[28–30]. α-cells express the somatostatin receptor SSTR2, which leads to hyperpolarization via activation of GIRK-channels[31,32]. Somatostatin also inhibits the exocytosis machinery via calcineurin[32], and inhibits α-cell exocytosis by effectively decreasing cytosolic cAMP[33,34]. Insulin receptor signaling is required for the suppression of glucagon secretion in vivo[35], but the precise mechanisms behind this are still debated[20,34,36]. Decreased sensitivity to insulin (or somatostatin) may therefore underlie the inadequate glucagon secretion in type-2 diabetes[37].

Glucagon is stored in ~7000 granules (diameter ~ 270 nm) and secreted by $Ca^{2+}$- and SNARE protein-dependent exocytosis[38,39]. At any time, only ~1% of these granules are in a releasable state that can undergo exocytosis upon $Ca^{2+}$-influx[40]. Paracrine signaling and glucose regulate glucagon secretion at least in part by affecting the size of this releasable pool of granules[32,38,41]. In many endocrine cells, secretory granules become release ready by sequential docking at the plasma membrane and assembly of the secretory machinery (priming)[42,43]. Although disturbances in these steps have been documented in β-cells of type-2 diabetic donors[44–47], they have not yet been studied in α-cells. An obstacle for understanding the regulation of α-cells has been the difficulty to isolate intrinsic and paracrine factors of α-cell regulation, as well as species differences between humans and rodent models. Glucagon secretion in vivo and in intact islets is affected by the presence of neighboring cell types, while single-cell electrophysiological measurements are invasive and may not reflect the in vivo situation.

In the current work, we took an optical approach to study glucagon granule exocytosis in α-cells of non-diabetic (ND) and type-2 diabetic (T2D) human subjects. We report that α-cells within intact islets respond with physiological inhibition of exocytosis by elevated glucose, whereas dispersed α-cells have a V-shaped response to glucose due to the lack of paracrine inhibition by insulin and somatostatin from neighboring β- and δ-cells. Importantly, α-cells of T2D are resistant to inhibition by insulin and somatostatin, which might underlie the hyperglucagonemia in type-2 diabetes.

## Results

### Exocytosis of glucagon granules in human α-cells.
Docking and exocytosis of glucagon granules at the plasma membrane was studied in dispersed islet preparations from 68 non-diabetic (ND) donors that all had glycated hemoglobin HbA1c values <6% (average 5.57 ± 0.29%, Supplementary Fig. 5). To identify α-cells, we transduced with Pppg-EGFP and the secretory granule marker NPY-mCherry, or with Pppg-NPY-EGFP (Fig. 1a and S1a, b) to drive expression of fluorescent proteins from the pre-proglucagon promoter (see methods). After culture for 26–48 h, α-cells were imaged by total internal reflection (TIRF) microscopy, which selectively images fluorescence near the plasma membrane (exponential decay constant $\tau \sim 0.1$ μm). The granule marker had a punctate staining pattern and excellent overlap with anti-glucagon immunostaining (Fig. 1a). Local application of elevated $K^+$ (75 mM, replacing $Na^+$) to depolarize the cells resulted in exocytosis, seen as rapid disappearance of individual fluorescently labeled granules (gr, see Fig. 1b and examples in S1B,C). Exposure to elevated $K^+$ for 40 s released 0.078 ± 0.004 granules μm$^{-12}$ (169 α-cells/29 ND donors, Fig. 1c, black). Exocytosis proceeded initially with a burst ($5.2 \times 10^{-3}$ gr μm$^{-2}$ s$^{-1}$ during the first 10 s) and decreased later to <0.6 × 10$^{-3}$ gr μm$^{-2}$ s$^{-1}$; these rates are about one-third of those observed in human β-cells[44]. Fitting the cumulative exocytosis ($n = 1530$ granules) with a double exponential function revealed two components with time constants of $\tau = 3.6 ± 0.2$ s and $19.9 ± 0.9$ s (Fig. 1c). The faster component made up 39 ± 3% of the total response, and likely corresponds to the RRP. Exocytosis occurred in granules that from the start of the experiment had been docked at the plasma membrane. We therefore quantified changes in docked granules during the experiment, by measuring the density of granules in the TIRF field. Stimulated exocytosis partially depleted docked granules (Fig. 1d), indicating that replacement by docking of new granules is relatively slow (a notion confirmed by a double stimulation protocol, Supplementary Fig. 2). On average, 13 ± 0.7% of the docked granules were released during the stimulation. Thus, depolarization of α-cells results in exocytosis with biphasic kinetics similar to those in other endocrine cells, indicating the existence of granule pools with differing release probabilities.

### Reduced granule docking and exocytosis glucagon in T2D α-cells.
During the course of this study, we also received islets from 21 donors that had been clinically diagnosed with type-2 diabetes (T2D), or whose glycated hemoglobin (HbA1c) values were above 6% (average 6.6 ± 0.7%, Supplementary Fig. 5). In dispersed T2D α-cells, $K^+$-stimulated exocytosis was reduced to 66 ± 8% of that in ND α-cells ($p = 0.004$, 0.052 ± 0.005 gr μm$^{-2}$; 75 cells/12 donors; Fig. 1b, c), mostly due to a reduced amplitude of the fast component ($\tau = 2.1 ± 0.1$ s, 25 ± 3%, $n = 441$ granules, $p = 0.0008$ vs ND). Docked granules were slightly fewer in T2D α-cells (0.57 ± 0.02 gr μm$^{-2}$, 106 cells/17 donors) compared with ND (0.61 ± 0.008 gr μm$^{-2}$; $p = 0.01$, 399 cells/50 donors, Fig. 1d). Exocytosis and granule density correlated on a per-donor basis (Pearson $r = 0.42$, $p = 0.006$, 41 donors; Fig. 1e), and both exocytosis ($r = 0.49$, $p = 0.002$; Fig. 1f) and docked granules ($r = 0.37$, $p = 0.003$; Fig. 1g) anti-correlated with the donor's HbA1c, as is the case in human β-cells[44]. The relationships are surprising given that reduced glucagon secretion should lead to reduced blood glucose and HbA1c values, and suggest that the diabetic state is causal for the reduced release capacity of T2D α-cells.

### Voltage-dependent currents are normal in T2D α-cells.
Since exocytosis in α-cells depends on $Ca^{2+}$-influx, we characterized voltage-dependent ion currents using patch-clamp

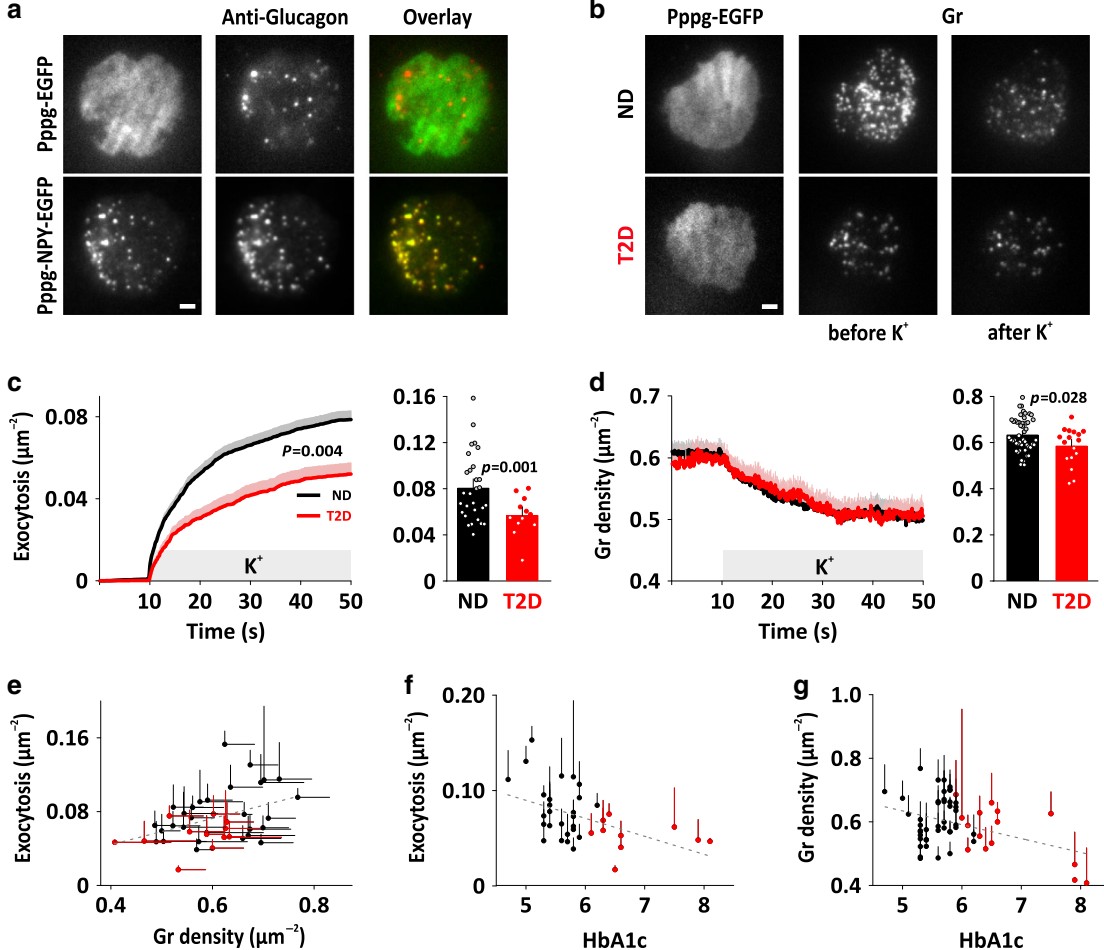

**Fig. 1 Exocytosis of glucagon granules in normal and diabetic pancreatic α-cells. a** TIRF images of dispersed human islet cells transduced with Pppg-EGFP (top) or Pppg-NPY-EGFP (bottom). In total, 90% of EGFP expressing cells ($n = 91$ cells, 5 donors) and 93% of NPY-EGFP expressing cells ($n = 70$ cells, 4 donors) were positive for glucagon. Scale bar 2 μm. **b** Examples of TIRF microscopy of cells from non-diabetic (ND) and type 2 diabetic (T2D) donors expressing Pppg-EGFP together with the granule marker NPY-mCherry (gr). Examples are before and after stimulation with 75 mM $K^+$ for 40 s. ($K^+$ was elevated during 10–50 s). Scale bar 2 μm. **c** Timecourse of average cumulative number of exocytotic events normalized to cell area in experiments as in **b** (left); 1530 granules in 169 ND cells (black), 441 granules in 75 T2D cells (red). Exocytosis (right) was $0.055 \pm 0.005$ gr μm$^{-2}$ in 12 T2D donors compared with $0.077 \pm 0.005$ gr μm$^{-2}$ in 29 ND donors ($p = 0.001$, two-tailed $t$-test). In **c**, **d**, timecourse (left) shows mean ± SEM of all cells, bargraphs (right) show donor means (dots) and mean ± SEM of individual donor means (bars). **d** Time courses of granule (gr) density (left) in ND or T2D cells as in **b**. Glucagon density (right) was $0.56 \pm 0.017$ gr μm$^{-2}$ in 17 T2D donors compared with $0.61 \pm 0.01$ gr μm$^{-2}$ in 50 ND donors ($p = 0.028$, two-tailed $t$-test). **e** Total exocytosis during $K^+$-stimulation plotted as function of granule density. Each symbol in **e**–**g** represents represent individual donors ± SEM (averages for $n = 29$ ND donors in black, and $n = 12$ T2D donors in red; 5–20 cells for each donor). Correlation was quantified as Pearson coefficient $r$ (see main text). **f** Total exocytosis during $K^+$-stimulation plotted as function of donor HbA1c. $n = 26$ ND donors and $n = 10$ T2D donors. **g** Granule density as function of donor HbA1c; $n = 40$ ND and $n = 15$ T2D donors.

electrophysiology. Dispersed ND or T2D α-cells were voltage clamped in whole-cell mode, and subjected to step depolarizations up to +70 mV from a holding potential of −70 mV (Fig. 2a). Analysis of the resulting inward currents revealed peak $Ca^{2+}$ (Fig. 2b) and $Na^+$-currents (Fig. 2c) that were of similar amplitude in ND and T2D cells. Half-maximal $Ca^{2+}$-current activation was reached at $-23 \pm 0.4$ (ND) and $-25 \pm 1.4$ mV (T2D, n.s.), and half-maximal $Na^+$-current activation was at $-25 \pm 0.6$ (ND) and $-24 \pm 0.8$ mV (T2D, n.s.). We also determined depolarization evoked membrane capacitance increases, a measure of exocytosis (Fig. 2d, e). A train of 14 depolarizations to 0 mV lasting 200 ms each resulted in a total capacitance increase of $112 \pm 19$ fF in ND cells and $78 \pm 19$ fF in T2D cells. This corresponds to a reduction of exocytosis by $25 \pm 10\%$ in T2D ($p = 0.1$), which is similar to the reduction observed by imaging granule release (Fig. 1d). Cell size, as assessed by cell

capacitance was not different in the two groups (Fig. 2f). Thus, reduced exocytosis in T2D α-cells cannot be explained by changes in $Ca^{2+}$-channel behavior.

**Glucose regulation of glucagon secretion.** Next, we determined the physiological glucose dependence of dispersed α-cells by measuring spontaneous exocytosis in a range of ambient glucose concentrations (1, 3, 7, 10, or 20 mM; at least 20 min pre-incubation), without imposing any depolarization. In movies lasting 3 min, we quantified granule exocytosis, docked granules, and the rate of docking of new granules at the plasma membrane (Fig. 3a–d). Spontaneous exocytosis was observed in all glucose concentrations, with a bimodal (V-shaped) dose response to glucose. Inhibition was about half at the nadir of 7 mM glucose, and ND and T2D cells behaved essentially identically (Fig. 3b).

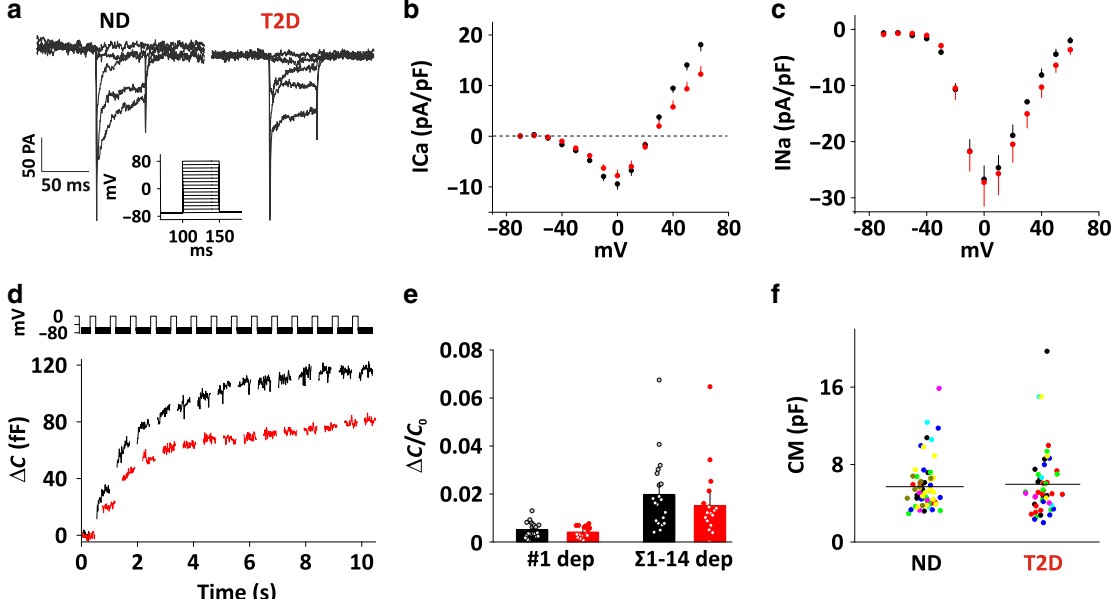

**Fig. 2 Voltage-dependent currents and exocytosis in human α-cells. a** Families of voltage-clamp current responses in human ND and T2D α-cells. Currents were elicited by 50 ms depolarizing pulses (−70 to +80 mV in 10 mV increments) from a holding potential of −70 mV. For clarity, only the responses between −40 mV and +10 are shown. **b, c** Current (I)–voltage (V) relationships for $Ca^{2+}$ (B, average current during 5–45 ms of the depolarization in **a**) and $Na^+$ (**c**, peak current during the first 5 ms of the depolarization in **a**) currents recorded from ND ($n = 38$, 4 donors, black) and diabetic T2D ($n = 32$, 3 donors, red) cells as in **a**. Currents are normalized to cell size (pF). Data are presented as mean values ± SEM. **d** Cell capacitance increase (ΔCm) during a train of 14 × 200-ms depolarizations from −70 mV to 0 mV in ND (black) and T2D (red) α-cells. **e** Average change in membrane capacitance, normalized to initial cell capacitance (ΔC/C₀), during the 1st depolarization (#1), and total increase during the train (Σ1-14) for ND ($n = 20$, 4 donors, black) and T2D ($n = 18$, 3 donors, red) α-cells. Data are presented as mean values ± SEM. **f** Whole-cell membrane capacitance (CM) in T2D ($n = 48$, 7 donors) and ND ($n = 66$, black, 8 donors) α-cells. Dots represent individual cells, and lines are mean values. Each donor is represented by a single color.

Docked granules, and to a lesser degree the rate of docking (in the same cells) likewise had a bimodal response to glucose, with a nadir at 7 mM (Fig. 3a, c, d). In separate experiments, we noticed that the response to changes in the glucose concentration was slow, and only minor during 2 min observation (Supplementary Fig. 3a, b). Thus, the glucose dependence of dispersed α-cells does not reflect physiological glucagon secretion, and exocytosis is accelerated in the hyperglycemic range instead of being inhibited. The data also suggest that the availability of docked granules is part of the regulation of exocytosis in α-cells.

To experimentally isolate direct glucose effects on the exocytosis machinery, we applied $K^+$-stimulations to α-cells bathed in 1, 7, or 10 mM glucose (Fig. 3e). This approach primarily elicits exocytosis of granules that are "primed" for exocytosis, the readily releasable pool (RRP). At all glucose concentrations, elevated $K^+$ stimulated biphasic exocytosis that by far exceeded spontaneous exocytosis (Fig. 3e). However, we observed again a bimodal glucose dependence with a nadir at 7 mM (38 cells/6 donors), where $K^+$-induced exocytosis was reduced by about half compared with 1 mM glucose (−45 ± 9%, $p = 8 \times 10^{-4}$, 30 cells/6 donors), or 10 mM glucose (−47 ± 6%, $p = 5 \times 10^{-6}$, 71 cells/14 donors). This reduction was most prominent during the initial burst phase, which may correspond to the immediately releasable pool of granules in β-cells[45]. At all glucose concentrations, $K^+$-stimulated exocytosis was significantly slower in T2D α-cells than in ND, but the bimodal glucose dependence of docking and exocytosis was preserved (Fig. 3f). We conclude that the α-cell exocytosis machinery is regulated by the ambient glucose concentration.

**Exocytosis in α-cells within intact islets.** The unexpected V-shaped glucose response of dispersed α-cells indicates that intrinsic regulation cannot explain the physiological inhibition of glucagon secretion in hyperglycemia. An alternative are paracrine effects from neighboring β- and δ-cells, which prompted us to quantify glucose dependent exocytosis in α-cells within intact islets (Fig. 4a, b). As expected, exocytosis of α-cells in intact islets varied with the glucose concentration, with inhibition in 7 mM glucose by 67 ± 15% ($p = 0.016$; 10 islets/3 donors), compared with 1 mM glucose (11 islets/3 donors). Exocytosis was also inhibited in 10 mM (by 56 ± 16% vs 1 mM, $p = 0.024$, 16 cells/4 donors), in contrast to dispersed α-cells. However, this inhibition was prevented by the SSTR2-specific somatostatin receptor antagonist CYN154806 (200 nM in the bath solution; $p = 0.003$ vs 10 mM glucose, 19 cells/3 donors). Similarly, block of insulin action with the insulin receptor antagonist S961 (1 μM) prevented inhibition of exocytosis at 10 mM glucose ($p = 0.003$ vs 10 mM glucose, 11 cells/2 donors), indicating that paracrine signaling is required for proper glucagon control in hyperglycemic conditions. In intact islets of T2D donors, α-cell exocytosis had a bimodal dose response to glucose that lacked inhibition hyperglycemia (Fig. 4c). At 7 mM glucose (16 islets/4 donors), exocytosis was inhibited to about half compared with 1 mM (21 islets/5 donors), whereas 10 mM glucose has no effect (29 islets/5 donors). Thus, α-cells within intact ND islets have a physiological response to glucose. Disruption of paracrine signaling by antagonists or islet dispersion leads to a V-shaped glucose response, similar to that observed in intact T2D islets.

We confirmed[48] expression of SSTR2 in human α-cells by co-immunostaining the receptor and glucagon in pancreatic sections of 10 human donors (5 ND, 5 T2D; Fig. 4d). In ND islets, SSTR2 distribution was mostly confined to the cell membrane of α- and other islet cells, whereas in T2D islets the SSTR2 staining was both weaker and largely vesicular (Fig. 4d). Quantitative analysis confirmed this conclusion and estimated that SSTR2 surface

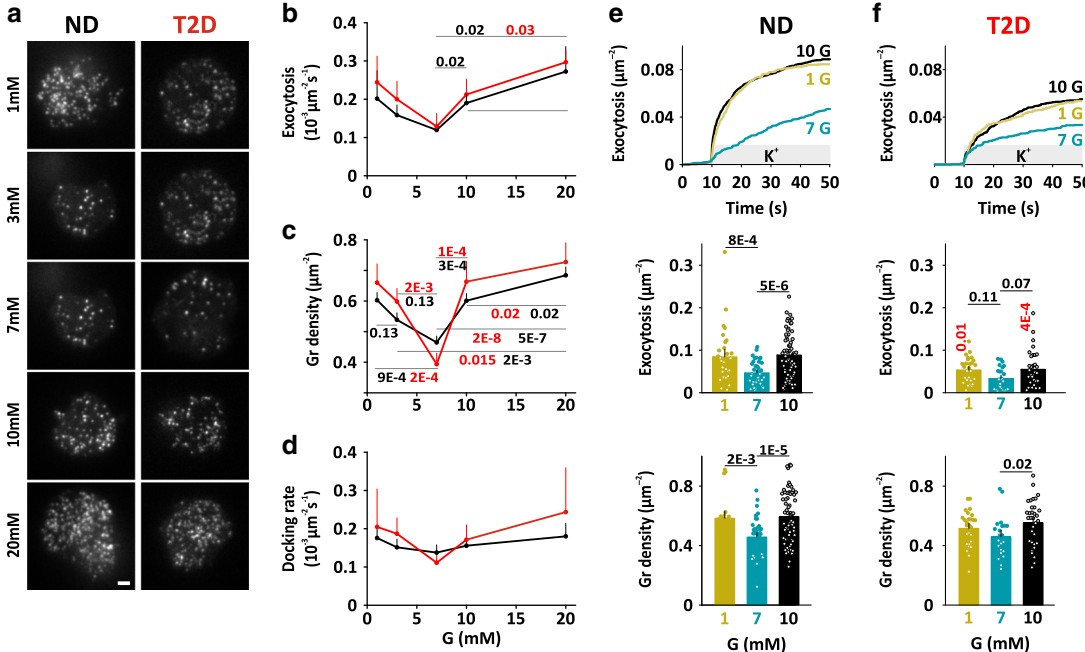

**Fig. 3 Glucose dependence of α-cell exocytosis. a** Examples TIRF images of dispersed ND (left) and T2D α-cells (right) expressing Pppg-NPY-EGFP, after equilibration in the indicated glucose concentrations. Scale bar 2 μm. **b** Average exocytosis as function of ambient glucose concentration for dispersed ND (black) and T2D (red) α-cells as in **a**. For ND, $n = 13$ cells/3 donors at 1 mM, 8 cells/2 donors at 3 mM, 13 cells/2 donors at 7 mM, 15 cells/3 donors at 10 mM, and 10 cells/2 donors at 20 mM). For T2D, $n = 7$ cells/2 donors at 1 mM), 9 cells/2 donors at 3 mM, 10 cells/2 donors at 7 mM, 10 cells/2 donors at 10 mM, and 7 cells/2 donors at 20 mM. Data are presented as mean values ± SEM. **c** Docked granules (average granule density) as function of ambient glucose concentration for ND (black, $n = 32$ cells/5 donors at 1 mM, 32 cells/5 donors at 3 mM, 32 cells/5 donors at 7 mM, 32 cells/5 donors at 10 mM, and 32 cells/5 donors at 20 mM) and for T2D cells (red, $n = 11$ cells/2 donors, 11 cells/2 donors at 3 mM, 17 cells/2 donors at 7 mM, 16 cells/3 donors at 10 mM, and 11 cells/2 donors at 20 mM). Data are presented as mean values ± SEM. In **b**, **c** p-values in black or red for comparisons as indicated by a line in ND group or T2D group respectively (two-tailed t-test). **d** Average rate of docking (granules becoming immobilized in the TIRF plane) as function of the ambient glucose concentration in the same cells as in **b**. Data are presented as mean values ± SEM. **e** Cumulative time course (upper), total exocytosis (middle), and initial density of docked granules (lower) during $K^+$-stimulated exocytosis in dispersed ND α-cells bathed in 1 mM (30 cells/6 donors, yellow), 7 mM (38 cells/6 donors, blue) or 10 mM glucose (71 cells/14 donors, black). Stimulation was from 10 to 50 s. Data are presented as mean values ± SEM. **f** As in **e**, but for dispersed T2D α-cells. $n = 27$ cells/5 donors at 1 mM, 24 cells/3 donors in 7 mM, and 33 cells/6 donors in 10 mM glucose. In **e**, **f**, p-values in black for comparisons as indicated by a line, or in red comparing ND and T2D for the same condition (oneway ANOVA, Fisher posthoc test).

expression is decreased by $44 \pm 7\%$, in T2D (824 cells/5 T2D donors vs 828 cells/5 ND donors Fig. 4e). Glucagon levels and distribution were similar in both groups (Fig. 4f). Insensitivity to somatostatin is therefore the result of excessive receptor internalization, as has recently been shown for pituitary cells[49].

**Paracrine regulation of exocytosis in dispersed α-cells**. Glucagon secretion is regulated by a network of paracrine mechanisms, some of which act directly on α-cells. We therefore quantified $K^+$-stimulated exocytosis dispersed α-cells in presence of a panel of islet paracrine effectors (somatostatin (SST, 400 nM), insulin (INS, 100 nM), forskolin (FSK, 2 μM), ϒ-aminobutyric acid GABA (400 nM), adrenaline (ADR, 5 μM), or glutamate (Glut, 1 mM), all present in the bath) at 1 or 10 mM glucose (Fig. 5). In 10 mM glucose (Fig. 5a), the δ-cell hormone somatostatin inhibited $K^+$-stimulated exocytosis by $65 \pm 4\%$ ($p = 2 \times 10^{-6}$, 53 cells/9 donors, vs control 71 cells/14 donors). β-cell factors likewise inhibited exocytosis, with insulin reducing it by $53 \pm 5\%$ ($p = 8 \times 10^{-5}$, 53 cells/8 donors) and GABA reducing it by $24 \pm 13\%$ (n.s., 14 cells/3 donors). In contrast, adrenaline doubled ($p = 4 \times 10^{-9}$, 30 cells/5 donors) and glutamate tripled α-cell exocytosis ($p = 14 \times 10^{-20}$, 16 cells/3 donors). Elevated cAMP, after exposure to forskolin, had no effect on exocytosis (n.s., 30 cells/5 donors), in contrast to previous reports[50]. In α-cells of T2D donors (Fig. 5b), adrenaline accelerated exocytosis about three fold ($p = 6 \times 10^{-9}$; 19 cells/3 donors). In contrast, the

inhibition by somatostatin or insulin was lost in T2D. None of the tested compounds affected the density of docked granules (Fig. 5a, b lower), suggesting that paracrine factors modulate α-cell exocytosis by affecting granule priming, rather than docking.

In hypoglycemic conditions (1 mM glucose, Fig. 5c, d), neither somatostatin (27 cells/6 donors, blue) nor insulin (24 cells/5 donors, green) affected $K^+$-stimulated exocytosis of dispersed α-cells, while adrenaline ($p = 2 \times 10^{-8}$, 15 cells/3 donors, pink) and glutamate ($p = 9 \times 10^{-10}$, 9 cells/2 donors, orange) accelerated exocytosis 2–3-fold compared with control (1 mM glucose, 30 cells/6 donors). T2D α-cells (Fig. 5d) behaved identical to ND α-cells with regard to somatostatin (17 cells/3 donors, blue), insulin (23 cells/3 donors, green) and adrenaline ($p = 2 \times 10^{-12}$, 10 cells/2 donors, pink), except for a moderate reduction of exocytosis at 1 mM glucose. No differences in the density of docked granules were observed in presence of any of the effectors, or comparing T2D with ND cells (Fig. 5c, d lower).

**Rapid paracrine inhibition by insulin and somatostatin.** Glucagon secretion oscillates with a frequency of minutes, which is inconsistent with the relatively slow glucose dependent regulation (Supplementary Fig 3). We therefore determined the time course of paracrine inhibition of spontaneous exocytosis by rapidly applying somatostatin (Fig. 6a, b, blue shading) or insulin (Fig. 6c, d, green shading) to dispersed α-cells. Maximal inhibition by somatostatin was reached within a few seconds (mono-

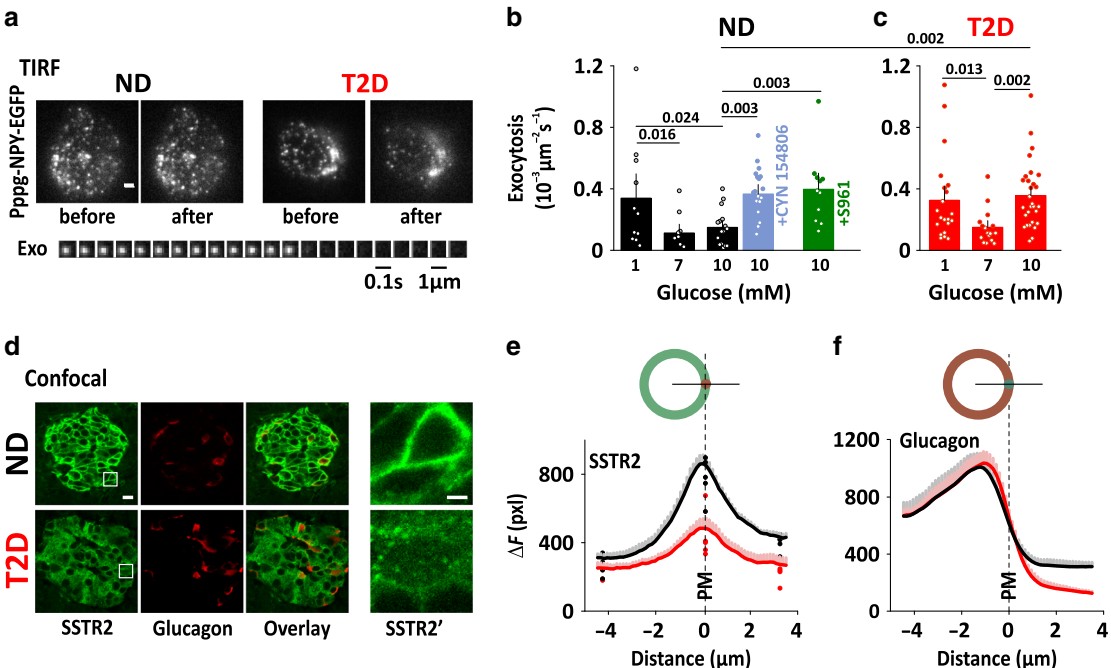

**Fig. 4 Disturbed paracrine signaling in α-cells within intact islets. a** TIRF images of NPY-EGFP of an α-cell within an intact islets of ND (left, representative for 67 cells) or T2D (right, representative for 66 cells) human donors. Scale bar 2 μm. Lower: image sequence of an exocytosis event in the ND α-cell example. **b** Average spontaneous exocytosis in α-cells within intact ND islets that were bathed in 1 mM (11 cells/3 donors, black), 7 mM (10 cells/3 donors, black), or 10 mM glucose (16 cells/4 donors, black), and in 10 mM glucose with SSTR antagonist (200 nM CYN154806, 19 cells/3 donors, light blue) or insulin receptor antagonist (1 μM S961, 11 cells/2 donors, green). Data are presented as mean values ± SEM. Scale bar 1 μm. **c** As in **b**, but for α-cells within intact T2D islets at 1 mM (21 cells/5 donors), 7 mM (16 cells/4 donors), and 10 mM glucose (29 cells/5 donors). In **b**, **c**, p-values are indicated for selected comparisons (one-way ANOVA with Fisher posthoc test). Data are presented as mean values ± SEM. **d** Representative confocal images of human pancreatic tissue sections of ND donors (top) and T2D donors (bottom), co-immunostained anti-SSTR2 (green) and anti-glucagon (red); scale bar 10 μm. The white square indicates the area that is enlarged in the right-most images (SSTR2; scale bar 2 μm). **e** Average SSTR2 staining intensity (F-background, average pixel value pxl), measured along a line across the plasma membrane of 828 cells from 5 ND donors (black) and 824 cells from 5 T2D donors (red). Cells were spatially aligned so that the line crosses the center-right of the cell perimeter at distance zero (illustrated drawing, top). Dots indicate average donor values. Staining intensities at distance zero were significantly different between ND and T2d donors ($p = 4 \times 10^{-11}$, two-tailed t test). **f** As in **b–e**, but for glucagon staining.

exponential decay constant $\tau = 9.4 \pm 5.3$s $n = 36$ cells, 6 donors). The effect frequently wore off towards the end of the 1 min challenge, and was rapidly reversed by removing the hormone (Fig. 6a). Similarly, rapid effects on exocytosis were observed with insulin (21 cells, 4 donors, Fig. 6c) or adrenaline (Supplementary Fig. 4). Thus, paracrine regulation of glucagon exocytosis occurs on a timescale that is consistent with the observed glucagon pulsatility[26]. Due to the small effect size, the time course of somatostatin or insulin inhibition of exocytosis could not be determined for T2D α-cells (Fig. 6b, d).

Finally, we tested the effect of somatostatin and insulin on α-cell electrical activity by perforated patch-clamp electrophysiology (Fig. 6e, h). In 10 mM glucose, electrical activity consisted action potential trains as described previously[38,51]. In ND α-cells, pulses of somatostatin (400 nM) caused rapid cessation of electrical activity (Fig. 6e). In the example in Fig. 6e top, action potentials reappeared ~20 s after removal of the somatostatin. However, in many cells electrical activity reappeared already during the somatostatin pulse, which is apparent the averaged membrane potential (Fig. 6e, bottom). Similar recovery during the somatostatin pulse was also seen for exocytosis (Fig. 6a), and may reflect somatostatin receptor inactivation. Insulin also dampened electrical activity, but these effects were both slower and weaker than those of somatostatin (Fig. 6f). In α-cells from T2D donors, the somatostatin or insulin pulses had little effect on the time course of exocytosis (Fig. 6b, d) or electrical activity (Fig. 6g, h). In summary, insulin and somatostatin inhibit both exocytosis (Fig. 5) and electrical activity

(Fig. 6). Both effects are lost in T2D, which is consistent with the notion that these cells are resistant to paracrine inhibition.

## Discussion

Glucose controls glucagon secretion by intrinsic and paracrine mechanisms, but their relative significance is still debated[52], and secretory defects in type-2 diabetes are not well understood. The current work is first in using high-resolution microscopy to study glucagon secretion both in intact islets and in single dispersed α-cells of healthy and type-2 diabetic donors, thus isolating intrinsic from paracrine mechanisms while having full control over paracrine signaling. We show that in the absence of paracrine influence, isolated α-cells respond appropriately to hypoglycemia with an increase in glucagon granule exocytosis. This is consistent with the glucose dependence of glucagon secretion from intact islets (but not FACS sorted α-cells)[52,53], and indicates that glucagon secretion in the lower glucose-concentration range is mostly under intrinsic control. With only 2-fold difference in the exocytosis rate between minimal secretion at 7 mM and maximal secretion at 1 mM glucose, the dynamic range is small compared with β-cells. Surprisingly, exocytosis of dispersed α-cells is stimulated in the hyperglycemic range, leading to an unphysiological V-shaped response with maximal exocytosis above 10 mM glucose. This is in contrast to intact islets, in which glucagon secretion is depressed between 3 and 20 mM glucose[54]. We confirm this here by exocytosis measurements in intact islets,

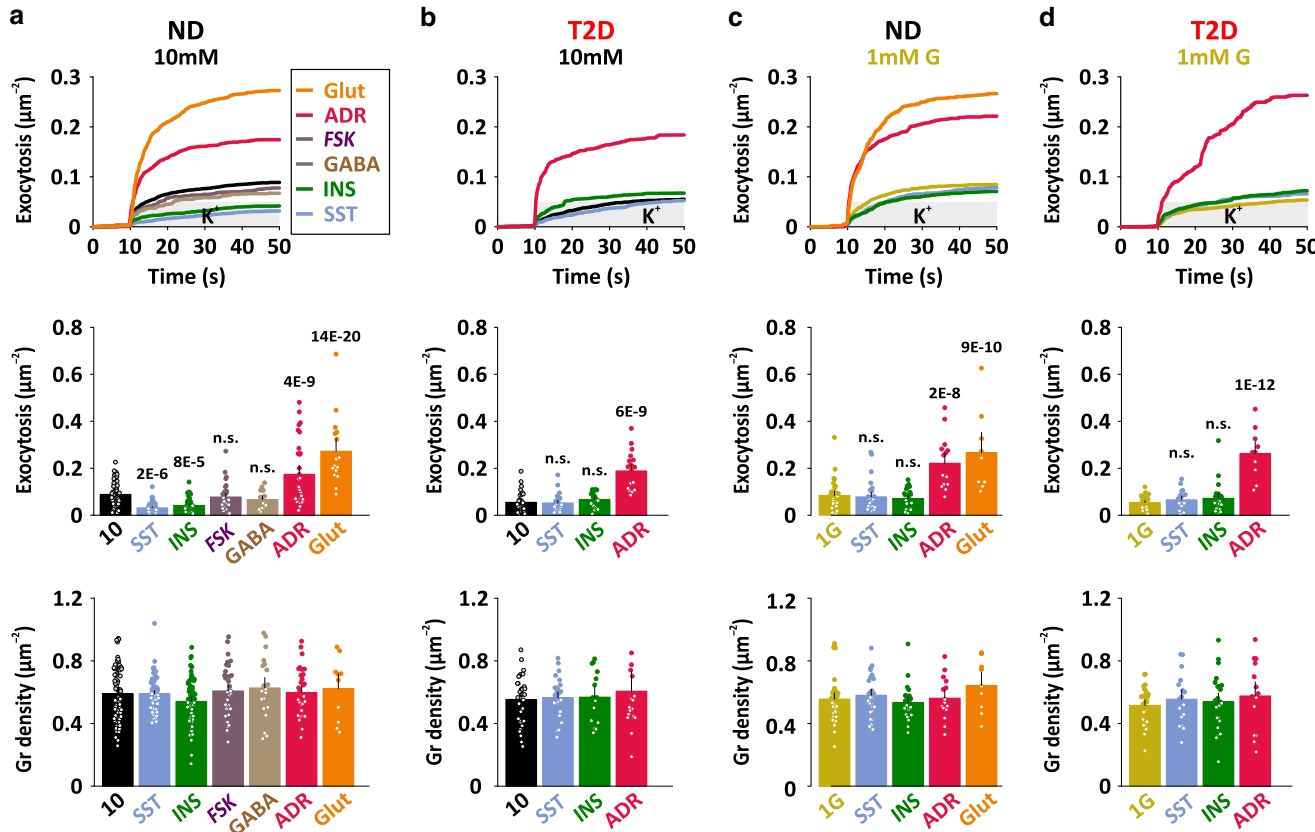

**Fig. 5 Paracrine regulation of exocytosis in dispersed α-cells. a** Cumulative time course (upper), total exocytosis (middle), and initial density of docked granules (lower) during K+-stimulated (gray bar) exocytosis in dispersed ND α-cells in control conditions (black, 10 mM glucose, $n = 71$ cells/14 donors) or exposed to somatostatin (light blue, SST, 400 nM, $n = 53$ cells/9 donors), insulin (green, INS, 100 nM, $n = 53$ cells/8 donors), forskolin (purple, FSK, 2 μM, $n = 30$ cells/5 donors), GABA (brown, 400 nM, $n = 14$ cells/3 donors), adrenaline (pink, ADR, 5 μM, $n = 30$ cells/5 donors), glutamate (orange, Glut, 1 mM, $n = 16$ cells/3 donors). In **a–d**, significant differences compared with control are indicated with $p$-values (one-way ANOVA, Fisher posthoc test). Data are presented as mean values ±SEM. **b** As in A, but for dispersed T2D α-cells. T2D ctrl $n = 33$ cells/6 donors, T2D SST $n = 26$ cells/5 donors, T2D INS $n = 19$ cells/4 donors, T2D ADR $n = 19$ cells/3 donors. **c, d** As in **a, b**, but in presence of 1 mM glucose. ND ctrl $n = 30$ cells/6 donors, ND SST $n = 27$ cells/6 donors, ND INS $n = 24$ cells/5 donors, ND ADR $n = 15$ cells/3 donors, ND Glut $n = 9$ cells/2 donors, T2D ctrl $n = 27$ cells/5 donors, T2D SST $n = 17$ cells/3 donors, T2D INS $n = 23$ cells/3 donors, and T2D ADR $n = 10$ cells/2 donors.

and provide evidence that this depression depends on the inhibitory effects on insulin and somatostatin that are released by neighboring β- and δ-cells. Evidently, α-cell intrinsic mechanisms are sufficient for the regulation of glucagon secretion hypo- and normoglycemic range (0–7 mM glucose), while paracrine inhibition is responsible for the physiological response in the hyperglycemic range. Consequently, appropriate glucagon secretion in the hyperglycemic range is lost when α-cells are removed from their context within the islet.

Elevated glucagon is a hallmark of type-2 diabetes[55]. Despite this, both glucose-dependent and depolarization (K+)-induced exocytosis was reduced in α-cells from donors that had been diagnosed with T2D. Both exocytosis and docked granules were moderately anti-correlated with donor HbA1c values. This indicates that the exocytosis machinery in α-cells from type-2 diabetics is slightly impaired, while electrophysiological parameters (that determine electrical activity, depolarization and Ca2+-influx) were normal. The reason for this is unknown, but may reflect reduced expression of certain exocytosis-related proteins, as is the case in β-cells[44,46,47]. The reduced exocytotic capacity in T2D α-cells is unrelated to changes in electrical activity, because it could be observed in K+-stimulation experiments in which the membrane potential is clamped. Since exocytosis in single α-cells is impaired rather than increased in T2D, the hyperglucagonemia in diabetic humans must be due to

mechanisms that are lost in isolated cells, such as paracrine or neuronal regulation[56]. In addition, gut derived glucagon may contribute to hyperglucagonemia following oral glucose intake[57,58].

Strikingly, the inhibitory effects of insulin and somatostatin on glucagon exocytosis were strongly reduced in cells from T2D donors, in parallel with internalization and reduced surface expression of SSTR2, the major somatostatin receptor in human α-cells. This points to α-cell resistance to insulin and somatostatin as the main cause for inadequate glucagon secretion in type-2 diabetes, which in turn exacerbates hyperglycemia[59]. Insulin resistance is a hallmark of T2DM, and has previously been proposed as mechanism for hyperglucagonemia. For example, insulin resistance is associated with fasting glucagon levels[37], and this inverse relationship is lost in type-2 diabetes[28]. Interestingly, SSTR2 surface expression was also reduced in β-cells within T2D islets, suggesting that reduced somatostatin sensitivity may contribute also to increased insulin secretion, as observed early during the development of T2D. While there is reason to believe that this is a consequence of altered δ-cells activity[60], its role may be to adapt islets to periods of greater food availability.

Exposure to insulin, somatostatin, and GABA reduced α-cell exocytosis, while adrenalin and glutamate stimulated it. This is consistent with the known effects of these signaling molecules on islets, as well as systemically[52]. We show here that these effects are

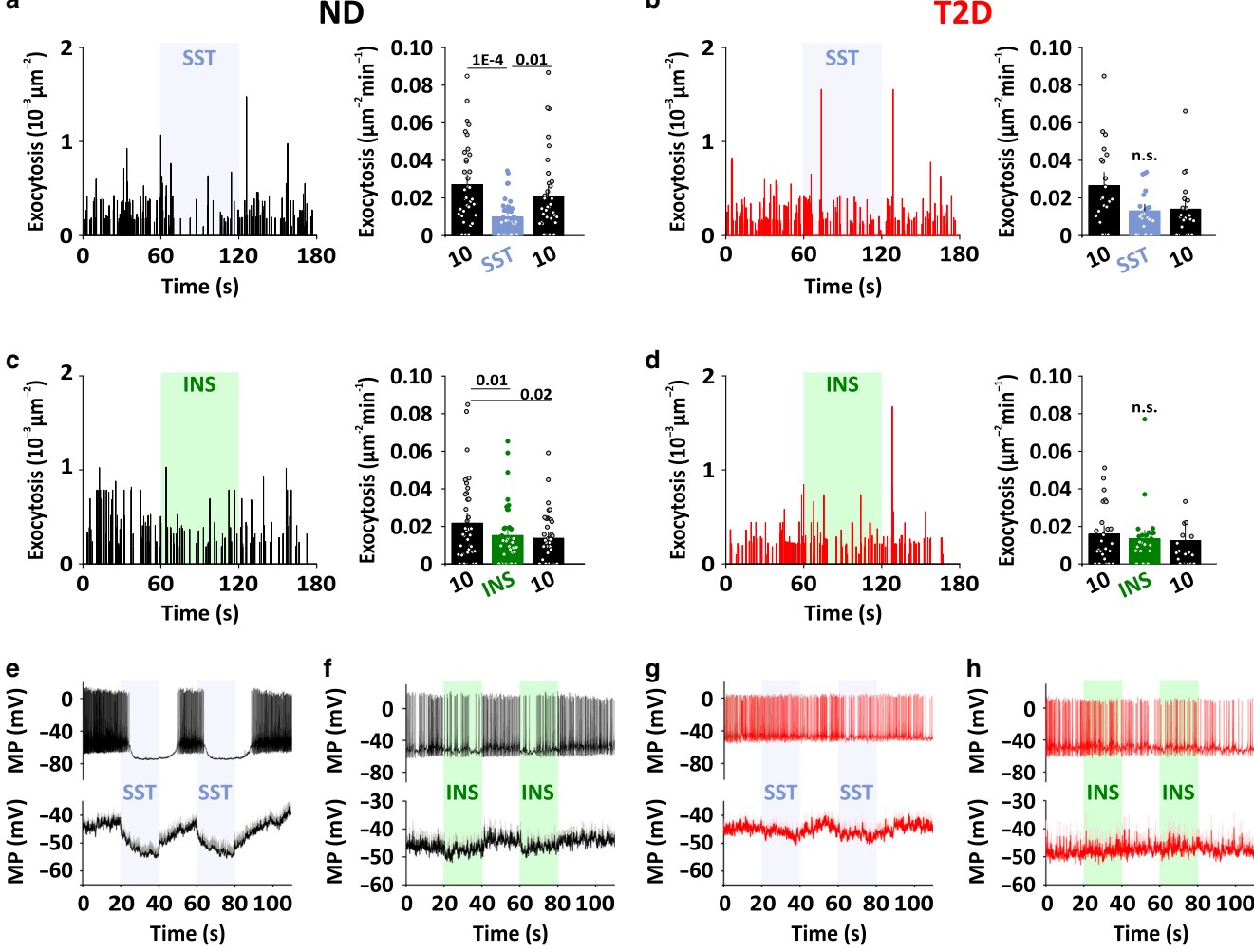

**Fig. 6 Time course of paracrine inhibition in dispersed α-cells. a** Time course of spontaneous exocytosis for a representative ND α-cell bathed in 10 mM glucose (black) and challenged with somatostatin during the indicated interval (SST, 400 nM; left, blue bar). Bars to the right show quantification of average exocytosis during the three time periods of the experiment (36 cells/6 donors). Data are presented as mean values ±SEM. In **a–d**, significant differences are indicated with p-values (one-way ANOVA, Fisher posthoc test). **b** As in **a**, but for T2D α-cells (38 cells/6 donors). **c**, **d** As in **b–e**, but challenged with insulin (INS, 100 nM, green bar). 21 cells/4 ND donors and 25 cells/3 T2D donors. **e**, **f** Example (upper) and average (lower) membrane potential recording in dispersed ND α-cells bathed in 10 mM glucose. Somatostatin (SST, 400 nM, 17 cells/4 donors, blue shading in **e**) or insulin (INS, 100 nM, 25 cells/6 donors, green shading in **f**) were applied during the indicated time interval. **g**, **h** as in **e**, **f**, but for dispersed T2D α-cells (11 cells/2 donors in G; 13 cells/4 donors in **h**).

very rapid (seconds), which is consistent with the frequency of pulsatile glucagon release in vivo and from intact islets. In the absence of glucose-dependent control, paracrine inhibition by insulin and somatostatin is therefore the most likely mechanism for glucagon regulation in the hyperglycemic range. It can be speculated that the differential glucose dependence of insulin and somatostatin secretion is reflected in different target glucose ranges for their action on α-cells. All tested paracrine modulators affected exocytosis machinery at the priming step, rather than by increasing granule docking. This is consistent with previous findings that somatostatin inhibits exocytosis in rat α-cells through $G_i$-dependent depriming[32], and reports that antagonists of SSTR2[12] or the associated G-protein cascade[12] increase glucagon secretion without altering the glucose-dependent inhibition of glucagon secretion. We did not observe any bursts of exocytosis, as might be expected given the pulsatile glucagon secretion from intact islets. This is in line with the absence of membrane potential oscillations in single cells[61] (that we confirm here), and indicates that the islet context is required not just for intra-islet synchronization, but for oscillatory α-cell behavior as such.

Capacitance measurements indicate that glucagon granules exist in at least two states with different release probabilities, which are often referred to as the readily releasable pool of granules (RRP) and a larger reserve pool (RP)[38,62]. We show here that glucagon granules were present at the plasma membrane for extended periods before undergoing exocytosis. We interpret this as the relatively slow conversion from RP to RRP that reflects the molecular assembly of the secretory machinery at the release site, in analogy with the situation in β-cells[42,44]. Throughout the glucose range, the rate of exocytosis was nearly identical to that of granule docking (Fig. 1b–d), suggesting that docking is rate limiting for secretion. This may indeed be the case during strong (non-physiological) stimulation, as illustrated by the finding that the glucose-dependence of depolarization-induced exocytosis followed that of granule docking. However, in physiological conditions elevated $K^+$ accelerated exocytosis ~50-fold (during the first second), which indicates a large excess in exocytotic capacity that is not triggered by normal α-cell electrical activity. A possible explanation could be that only a limited number of granules is positionally primed, i.e. located near voltage-gated $Ca^{2+}$- granules[63]. Further theoretical

work is required to understand the combination of factors affecting granule exocytosis, and the granule conversion rates provided here may be useful in this regard.

## Methods

**Tissue**. Pancreatic islets and pancreas sections were obtained from human cadaveric donors by the Nordic Network for Clinical Islet Transplantation Uppsala[64] (ethical approval by Uppsala Regional Ethics Board) or the ADI Isletcore at the University of Alberta[65] (ethical approval by Alberta Human Research Ethics Board, Pro00001754), with written donor and family consent for use in research. Work with human tissue complied with all relevant ethical regulations for use of human tissue in research and the study was approved by the Uppsala Regional Ethics Board (2006/348). Isolated islets were cultured free-floating in sterile dishes in CMRL 1066 culture medium containing 5.5 mM glucose, 10% fetal calf serum (FCS), 2 mM L-glutamine, streptomycin (100 U/ml), and penicillin (100 U/ml) at 37 °C in an atmosphere of 5% $CO_2$ up to 2 weeks. Islets were dispersed into single cells by gentle pipetting in cell dissociation buffer (Thermo Fisher Scientific) supplemented with trypsin (0.005%, Life Technologies). Cells were then washed and plated in serum-containing medium onto 22-mm polylysine-coated coverslips, allowed to settle overnight, and then transduced using adenovirus. In Fig. 4a, c, the dispersion step was omitted and intact islets were transduced with Pppg-NPY-EGFP adenovirus and allowed to settle onto 22-mm polylysine-coated coverslips.

**Labeling of human pancreatic α-cells and glucagon granules**. To identify α-cells, we transduced cells with adenovirus coding for enhanced green fluorescent protein (EGFP) under the control of the pre-proglucagon promoter[66]. The system takes advantage of Tet-On conditional expression in the presence of 4 μM doxycycline, to drive expression of EGFP. The cells were simultaneously transduced with adNPY-mCherry, a well-established secretory granule marker. Alternatively, adenovirus coding for EGFP-tagged neuropeptide Y under control of the pre-proglucagon promoter (Pppg-NPY-EGFP) was used, thus combining cell type identification and secretory granule label. For both approaches, immunostaining with an anti-glucagon antibody confirmed that over 90% of the fluorescently labeled cells were α-cells (Fig. 1a and Supplementary Fig 1a). Approximately one-third of the glucagon positive cells were labeled with Pppg-NPY-EGFP (Supplementary Fig 1a). NPY-EGFP labeled granules had excellent overlap with punctate glucagon staining (Fig. 1a), with 94 ± 1% of glucagon positive granules being labeled with NPY-EGFP (37 cells, 5 donors). We verified that exocytosis in the identified cells was stimulated by adrenaline (Supplementary Fig 1b–d), which increases intracellular $Ca^{2+}$ in α- but not β-cells.

**TIRF microscopy**. Cells were imaged using a custom-built lens-type total internal reflection (TIRF) microscope based on an AxioObserver Z1 with a ×100/1.45 objective (Carl Zeiss). Excitation was from two DPSS lasers at 491 and 561 nm (Cobolt) passed through a cleanup filter (zet405/488/561/640x, Chroma) and controlled with an acousto-optical tunable filter (AA-Opto). Excitation and emission light were separated using a beamsplitter (ZT405/488/561/640rpc, Chroma). The emission light was chromatically separated onto separate areas of an EMCCD camera (Roper QuantEM 512SC) using an image splitter (Optical Insights) with a cutoff at 565 nm (565dcxr, Chroma) and emission filters (ET525/50m and 600/50m, Chroma). Scaling was 160 nm per pixel.

Confocal microscopy was done with a Zeiss LSM780 using a 63/1.40 objective (Zeiss) with sequential scanning of the red (excitation 561 nm, emission 578–696 nm) and green channel (excitation 488 nm, emission 493–574 nm). Pinhole size was 0.61 mm, corresponding to 1 Airy unit.

Cells were imaged in a standard solution containing (in mM) 138 NaCl, 5.6 KCl, 1.2 $MgCl_2$, 2.6 $CaCl_2$, 10 D-glucose, 5 HEPES (pH 7.4 with NaOH). In Figs. 3–5 the D-glucose concentration was varied as indicated and cells equilibrated for at least 20 min before recording commenced. Where stated, exocytosis was evoked by elevating $K^+$ to rapidly depolarize the cells (75 mM KCl equimolarly replacing NaCl in the standard solution). In these experiments, $K^+$ was elevated from 10 s after the onset of the recording until the end of the experiment at 50 s. Timed applications of elevated $K^+$ (Figs. 1, 3e, f, 4, 5d black, and 6), adrenalin (Supplementary Figs. 1d and 4), insulin or somatostatin (in Fig. 6), and glucose changes (Supplementary Fig. 3) were by computer-controlled local pressure ejection from a pulled glass pipette (similar to those used for patch clamp). Spontaneous glucose-dependent exocytosis (Figs. 3a–d, 4a–c, and 6) was recorded for 3 min per after equilibration in the stated conditions. In Fig. 6, insulin or somatostatin were applied during the indicated times.

**Image analysis**. Exocytosis events were identified based on the characteristic rapid loss of the granule marker fluorescence (1–2 frames). Granule docking events were rare and defined as granules that approached the TIRF field and becoming laterally confined once they reached their maximum brightness[42].

Docked granules were counted using the 'find maxima' function in ImageJ (http://rsbweb.nih.gov/ij). Values were normalized to each cells' contact area with the coverslip (footprint). The $\Delta F$ parameter estimates the fluorescence that is specifically localized to a granule, but subtracting a local background value (average of a 5 pixel wide annulus) from the average fluorescence value in a 3 pixel wide circle, both centered at the granule position.

**Immunostaining of pancreatic sections**. For analysis of SSTR2 expression, deparaffinized human pancreatic tissue sections (biobank samples obtained from the EXODIAB consortium, Uppsala) were heated in a buffer containing 10 mM Tri-sodium citrate and 0.05% Tween 20 (pH 6) for 15 min, allowed to cool, and rinsed with wash buffer 1x. After a 30-min blocking step (Background Sniper, Biocare Medical), sections were rinsed with wash buffer 1x (Dako) and incubated with anti-SSTR2 (Abcam ab134152, diluted 1:500 in wash buffer), and anti-glucagon antibodies (Dako A0565, diluted 1:1500 in wash buffer) overnight at 4 °C. The slides were then washed in wash buffer and incubated with fluorophore-labeled secondary antibodies (diluted in Dako wash buffer 1x) for 30 min at room temperature. Fluorescence was visualized using a Zeiss LSM 780 confocal microscope. For analysis, 3-pixel wide linescans of fluorescent intensity were calculated as illustrated in Fig. 3e, f, top. Background subtracted and estimated as the minimum value to the left of the alignment point (corresponding to the nucleus location), after 3 × 3 median filtering.

**Electrophysiology**. Standard whole-cell voltage clamp and capacitance recordings were performed using an EPC-9 patch amplifier (HEKA Electronics, Lambrecht/Pfalz, Germany) and PatchMaster software. Voltage-dependent currents were investigated using an IV-protocol, in which the membrane was depolarized from −70 mV to +80 mV (10 mV steps) during 50 ms each. Currents were compensated for capacitive transients and linear leak using a P/4 protocol. Exocytosis was detected as changes in cell capacitance using the lock-in module of Patchmaster (30 mV peak-to-peak with a frequency of 1 kHz).

Patch electrodes were made from borosilicate glass capillaries coated with Sylgard close to the tips and fire-polished. The pipette resistance ranged between 2 and 4 MΩ when filled with the intracellular solution containing (in mM) 125 Cs-glutamate, 10 CsCl, 10 NaCl, 1 $MgCl_2$, 0.05 EGTA, 3 Mg-ATP, 0.1 cAMP, and 5 HEPES, pH 7.2 adjusted using CsOH.

During the experiments, the cells were continuously superfused with an extracellular solution containing (in mM) 138 NaCl, 5.6 KCl, 1.2 $MgCl_2$, 2.6 $CaCl_2$, 10 D-glucose, and 5 Hepes, pH 7.4 adjusted with NaOH at a rate of 0.4 ml/min. All electrophysiological measurements were performed at 32C. In the analysis of the measured voltage-dependent current consists of both $Na^+$ and $Ca^{2+}$ current components, were the rapid peak current (0–3 ms) represent the $Na^+$ current and the sustained current during the latter part of the depolarization reflects the $Ca^{2+}$ current. In Fig. 4, for membrane potential, solution was containing (in mM) 76 $K_2SO_4$, 10 KCl, 1 $MgCl_2$, and 5 HEPES, pH 7.3 adjusted with KOH.

**Statistics**. Data are presented as mean ± SEM unless otherwise stated. Statistical significance was tested using t-test for comparing ND and T2D groups, or ANOVA for multiple comparisons, as stated (in Origin 2018). Correlation was quantified as Pearson coefficient r using Excel.

**Reporting summary**. Further information on research design is available in the Nature Research Reporting Summary linked to this article.

## Data availability

The datasets generated during and/or analyzed during the current study are available from the corresponding author on reasonable request. A source data file containing all numeric analysis for Figs. 1c–g, 2b–e, 3b–f, 4b–f, 5a–d, and 6a–h, and Supplementary Figs. 1d, 2b–d, 3a, b, and 4a is provided with the paper.

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

## Acknowledgements

We thank Jan Saras for expert technical assistance, and Hongyan Shuai and Yunjian Xu for preparing Pppg-EGFP plasmid and adenovirus. Human islets were provided through the JDRF award 31-2008-416 (ECIT Islet for Basic Research Program) and the Alberta Diabetes Institute Islet-Core with assistance of the Human Organ Procurement and Exchange [HOPE] program, Trillium Gift of Life Network [TGLN], and other Canadian organ procurement organizations. The work was supported by the Swedish Science Council, Diabetes Wellness Network Sweden, the Swedish Diabetes Society, the European Foundation for the Study of Diabetes, the NovoNordisk Foundation, Excellence of Diabetes Research in Sweden (EXODIAB), and the Family Ernfors- and OE&E Johanssons-Foundations. Open access funding provided by Uppsala University.

## Author contributions

M.O.H. and S.B. designed experiments and analyzed the data. M.O.H. performed experiments. P.E.L. performed electro-physiology experiments. M.O.H. and N.R.G. prepared human islets for imaging. M.O.H. designed and generated the Pppg-NPY-GFP virus construct. A.T. supplied Pppg-GFP plasmid and adenovirus. S.B. conceived the study. S.B. and M.O.H. wrote, and all authors critically reviewed the manuscript.

## Competing interests

The authors declare no competing interests.
