## [Peer Review File · Nature Communications]

Reviewers' Comments:

Reviewer #1:

Remarks to the Author:

These workers, employing electrophysiology and high resolution TRIF microscopy, assessed the differences in intrinsic and paracrine regulation effects on glucagon granule exocytotic behavior of dispersed human islet alpha-cells from healthy and T2D donors. They concluded that glucose stimulated alpha-cell glucagon secretion is normally inhibited by paracrine insulin, somatostatin, and GABA. However, in T2D alpha-cells, this paracrine regulation by insulin is absent and with somatostatin it is blunted, and since these are dispersed single cells studies, this would suggest an intrinsic defect of a loss of sensitivity to insulin and somatostatin. This would in part explain why in T2D glucagon secretion is not inhibited thus the clinically observed hyperglucagonemia that contributes to the hyperglycemia in T2D. What is also new that was shown in this work is that alpha cell exocytosis is somewhat reduced in T2D. All of these effects seem to be due to defective priming of glucagon granule rather than docking, although the precise priming mechanism or identification of priming proteins for the glucagon granule exocytotic machinery was not assessed per se.

1. A deficiency in this work is that most of the stimulatory protocol was with supraphysiologic high K^+ and not solely on the varying glucose concentrations. With the high K^+ , it seems not surprising that a burst of exocytosis over a short term was observed, but not a presumably physiologic more prolonged pattern of exocytosis. Their explanation is that this may be due to slow recovering or the RRP, but this was not critically assessed which they could have done by electrophysiology.

2. While the strategy of using two modalities of electrophysiology and TIRF microscopy imaging are elegant, it would have been further informative and more rigorous to have included some morphologic EM studies to show total number of glucagon granules and how they are distributed across the alpha cell including whether there might be reduced number of docked granules. Perhaps the observed reduced priming could be in part due to reduced replenishment (less granule closed to the plasma membrane thus suggesting a possible problem with granule maturation or transport) rather than reduced priming of already docked granules.

3. Since the identity of alpha-cell is crucial to the entire study, these workers should have provided unequivocal data on the cell identities and not just rely on a previous published report, which was shown for PPPG-GFP but not PPPG-NPY-GFP. There should be more rigor for Figure 1A, B and S1A by showing negative controls. PPPG-GFP and NPY-GFP virus transduction of alpha cells ONLY should be demonstrated by insulin and somatostatin staining to demonstrate that there is absolutely no mis-targeting of other cell types. The infection efficiency was also not indicated – with the dispersed cells i.e. did both viruses infect 100% of alpha cells?

4. Some explanation is needed regarding with their electrophysiology studies.

(a) Half-maximal (V_h) Na^+ current activation potential was -25 mV in ND and -24 mV in T2D alpha-cells. However, the ND Na^+ V_h is different from that reported by Ramracheya et al, Diabetes 2010 (PMID 20547976), wherein their Na^+ V_h of healthy human alpha-cells averaged at -40 mV. Could the virus have inadvertent effects on some biophysical properties of alpha cells.

(b) In Figure 1M, cell sizes of alpha-cells in both ND and T2D groups ranged widely, so Figure 1K should be reanalyzed as ΔC_m (fF)/Cell size (pF) to ascertain their findings and conclusion about the extent of reduction in exocytosis capacity in the T2D alpha cells.

(c) Na^+ or Ca^{2+} channel currents are usually recorded in presence of $CoCl_2$ or TTX, respectively, to avoid contamination of each other and ensure accurate channel property analyses. This was not done in this paper.

(d) In Figure 4D, resting membrane potential was ~ 50 mV in T2D alpha-cells, which is more depolarized than ND alpha-cells (~ 70 mV, Fig 4C). Please comment. In Figure 4E,F, the Y-axes were labeled membrane potential (mV) but the figure legends state this as ?AP frequency. Please rectify or explain exactly.

Overall, this work using single cell recordings more critically analyzed the hypotheses regarding paracrine inhibition of alpha cells that were postulated by others but those studies had used less rigorous or precise strategies.

Reviewer #2:

Remarks to the Author:

In their study, Hmeadi, Barg and colleagues investigate how granule exocytosis and membrane excitability in isolated alpha cells is affected in type 2 diabetes. This is an important issue because it has been reported that in type 2 diabetes glucagon levels are unphysiologically high, thus contributing to hyperglycemia. Using exclusively human material, the authors employ sophisticated imaging and electrophysiological approaches to convincingly show that exocytosis is reduced in alpha cells from type 2 donors. Because these findings contradict the now widespread notion of diabetes-associated hyperglucagonemia, the authors attempt to demonstrate that there must be additional factors that could be responsible for increasing glucagon secretion in the diabetic state. Thus, they look at paracrine signaling from delta and beta cells. Here the experimental strategy relies on determining changes in the responses to exogenously applied signaling molecules such as somatostatin and insulin. As detailed below, this second part is less convincing. As a consequence, the results do not support the strong conclusion drawn in the manuscript that paracrine inhibition of alpha cells is disturbed in type 2 diabetes. An additional effort is required to be able to conclude that paracrine signaling is defective. Below is a list of suggestions and comments that could help improving this otherwise fine manuscript.

(1) The authors convincingly show that in type 2 diabetes glucose-dependent granule exocytosis is reduced in alpha cells. This is actually in line with a recent study showing that, contrary to widespread expectations, glucagon secretion is reduced in type 1 diabetes (Brissova et al., 2018, Cell Rep 22:2667). These are important results. It is still interesting (and also important) to study if paracrine interactions are affected in the islet in type 2 diabetes. However, as the authors probably will agree, paracrine signaling must be investigated differently and more thoroughly than in the manuscript in its current form. The experimental design used is not adequate because it only probes for receptor responses. It does not reveal endogenous paracrine activation (or lack thereof). What is required is to study these responses in intact islets and challenge intrinsic signaling with receptor antagonists or other pharmacological tools.

(2) If glucose-dependent glucagon secretion from intact islets were increased despite diminished exocytosis in isolated alpha cells, the authors could invoke other mechanisms such as loss of inhibitory paracrine signaling. I don't think that changes (e.g. increases) in glucagon secretion have been determined in islets from type 2 diabetic donors. This information is important for this paper.

(3) It would also be important to determine if receptor expression is downregulated. With many databases now available, this could be readily checked, and the data be included in the paper. By the way, a quick look at the Sandberg atlas does not show a reduction of somatostatin SSTR2 receptor expression in type 2 diabetes (<http://sandberg.cmb.ki.se/pancreas/>).

(4) The studies addressing paracrine signaling also require determining how insulin and somatostatin secretion change in the same type 2 preparations. In this context, it would be important to have an overarching hypothesis about what is going on. What are the putative mechanisms through which paracrine signaling is diminished in type 2 diabetes?

(5) More experimental detail is needed to be able to follow the figures. For instance, the authors should state how the concentrations for the different stimuli were selected, for how long the cells were incubated in the different glucose concentrations, and what statistical tests were used. It is

not clear from the Methods what protocols were used for the different studies. The use of the Student t-test is not adequate for multiple comparisons (see Figure 3d, e). In general, there is negligence with data analysis and presentation. This should be improved.

(6) In Figure 4e, f, the ordinate is wrongly labeled. It should be "action potential frequency", not "membrane potential".

(7) References 39 and 42 are the same.

Point by point response to Reviewers' comments:

We thank both reviewers for their time and effort in reviewing the manuscript!

Reviewer #1 (Remarks to the Author):

These workers, employing electrophysiology and high resolution TRIF microscopy, assessed the differences in intrinsic and paracrine regulation effects on glucagon granule exocytotic behavior of dispersed human islet alpha-cells from healthy and T2D donors. They concluded that glucose stimulated alpha-cell glucagon secretion is normally inhibited by paracrine insulin, somatostatin, and GABA. However, in T2D alpha-cells, this paracrine regulation by insulin is absent and with somatostatin it is blunted, and since these are dispersed single cells studies, this would suggest an intrinsic defect of a loss of sensitivity to insulin and somatostatin. This would in part explain why in T2D glucagon secretion is not inhibited thus the clinically observed hyperglucagonemia that contributes to the hyperglycemia in T2D. What is also new that was shown in this work is that alpha cell exocytosis is somewhat reduced in T2D. All of these effects seem to be due to defective priming of glucagon granule rather than docking, although the precise priming mechanism or identification of priming proteins for the glucagon granule exocytotic machinery was not assessed per se.

We would like to clarify that docked granules are reduced both in T2D, and at 7mM glucose. We believe that this contributes to the reduced exocytosis competence (both in T2D and at 7mM glucose) since reduced docked results in fewer granules that are available for priming. In contrast, paracrine factors act at the priming step and have little effect on docking.

1. A deficiency in this work is that most of the stimulatory protocol was with supraphysiologic high K⁺ and not solely on the varying glucose concentrations. With the high K⁺, it seems not surprising that a burst of exocytosis over a short term was observed, but not a presumably physiologic more prolonged pattern of exocytosis. Their explanation is that this may be due to slow recovering or the RRP, but this was not critically assessed which they could have done by electrophysiology.

This may have been a misunderstanding. We used glucose in Fig 2, 3A-D, 6, S1D red, S3, and S4), and the major conclusions of the paper are based on these data (in particular that of the V-shaped glucose dependence). In addition, we have now added data from intact data in Fig 4A-C. While this may not be apparent from the figures, these experimental series are major efforts due to the number of conditions and the long recordings required. As the reviewer notes, we also used K⁺-stimulation in a number of experiments. The K⁺ protocol is complimentary to the glucose experiments and recapitulates the instantaneous depolarization to 0mV that is often used in capacitance measurements (our method achieves this in about 50ms), to directly assess effects on the exocytosis machinery. Interestingly, we find very similar glucose dependence and paracrine inhibition in the two paradigms, indicating that exocytosis and electrical activity are regulated in parallel. The exocytosis reduction in T2D or at 7mM is mostly due to a reduction in the fast component of exocytosis (eg in Fig 3E-F upper), which we interpret as the RRP (as one would in capacitance measurements).

2. While the strategy of using two modalities of electrophysiology and TIRF microscopy imaging are elegant, it would have been further informative and more rigorous to have included some morphologic EM studies to show total number of glucagon granules and how they are distributed across the alpha cell including whether there might be reduced number of docked granules. Perhaps the observed reduced priming could be in part due to reduced replenishment (less granule closed to the plasma membrane thus suggesting a possible problem with granule maturation or transport) rather than reduced priming of already docked granules.

We used TIRF microscopy to quantify granule docking. Note that the “granule density” in TIRF images reflects granules at the plasma membrane and is commonly used to quantify docked granules. We have done this throughout the paper (Figs 1D,G, 3D-F, 5A-D lower, and S2B). In addition, we have quantified the rate at which new granules dock at the plasma membrane (Fig 3D). The two parameters are not necessarily the same, since the docked granule density is affected by the balance of docking, undocking, and ongoing exocytosis. An important conclusion of the paper is that docked granules are reduced at 7mM glucose, compared with 1 and 10mM, which likely contributes to the reduced exocytosis in this range. We would like to point out that the extremely low throughput of EM would have been prohibitive for this work.

3. Since the identity of alpha-cell is crucial to the entire study, these workers should have provided unequivocal data on the cell identities and not just rely on a previous published report, which was shown for PPPG-GFP but not PPPG-NPY-GFP. There should be more rigor for Figure 1A, B and S1A by showing negative controls. PPPG-GFP and NPY-GFP virus transduction of alpha cells ONLY should be demonstrated by insulin and somatostatin staining to demonstrate that there is absolutely no mis-targeting of other cell types. The infection efficiency was also not indicated – with the dispersed cells i.e. did both viruses infect 100% of alpha cells?

These experiments are now included (Fig S1A-B, in addition to the previously shown Fig 1A-B). The data confirm that only α -cells are stained, infection rate is about 1/3 of glucagon positive cells.

4. Some explanation is needed regarding with their electrophysiology studies.

(a) Half-maximal (V_h) Na^+ current activation potential was -25 mV in ND and -24mV in T2D alpha-cells. However, the ND Na^+ V_h is different from that reported by Ramracheya et al, Diabetes 2010 (PMID 20547976), wherein their Na^+ V_h of healthy human alpha-cells averaged at -40 mV. Could the virus have inadvertent effects on some biophysical properties of alpha cells.

We find it unlikely that biophysical properties were affected by the virus. One reason is that the electrical activity (arguably a rather sensitive measure of electrophysiological properties) is very similar in our work and in Ramracheya et al 2010. Moreover, the Na^+ -current IV relationship in that paper is near identical to the one we show in Fig 2C, despite the difference in reported V_h values. Both curves are left-shifted compared with data from mouse in Barg et al 2000.

(b) In Figure 1M, cell sizes of alpha-cells in both ND and T2D groups ranged widely, so Figure 1K should be reanalyzed as ΔC_m (fF)/Cell size(pF) to ascertain their findings and conclusion about the extent of reduction in exocytosis capacity in the T2D alpha cells.

The normalization is now included in Fig 2E. We are still unable to detect differences between ND and T2D.

(c) Na^+ - or Ca^{2+} channel currents are usually recorded in presence of $CoCl_2$ or TTX, respectively, to avoid contamination of each other and ensure accurate channel property analyses. This was not done in this paper.

This is true, but the two currents have very different activation and inactivation kinetics, which make them relatively simple to separate by setting time windows during the analysis. Since the focus was on identifying differences between NS and T2D we chose an approach that maximized throughput. Note that 70 cells were analyzed for the IV relationships, compared with only 5 cells in Ramracheya et al.

(d) In Figure 4D, resting membrane potential was ~50mV in T2D alpha-cells, which is more depolarized

than ND alpha-cells (~70mV, Fig 4C). Please comment. In Figure 4E,F, the Y-axes were labeled membrane potential (mV) but the figure legends state this as ?AP frequency. Please rectify or explain exactly.

Note that Fig 6E/G (was Fig 4D) shows examples, and we now include two additional examples (Fig 6F/H). The average membrane potential is indeed quite similar in ND and T2D.

The legend has been corrected (it should say “average membrane potential”). Thank you for pointing this out.

Overall, this work using single cell recordings more critically analyzed the hypotheses regarding paracrine inhibition of alpha cells that were postulated by others but those studies had used less rigorous or precise strategies.

Reviewer #2 (Remarks to the Author):

In their study, Hmeadi, Barg and colleagues investigate how granule exocytosis and membrane excitability in isolated alpha cells is affected in type 2 diabetes. This is an important issue because it has been reported that in type 2 diabetes glucagon levels are unphysiologically high, thus contributing to hyperglycemia. Using exclusively human material, the authors employ sophisticated imaging and electrophysiological approaches to convincingly show that exocytosis is reduced in alpha cells from type 2 donors. Because these findings contradict the now widespread notion of diabetes-associated hyperglucagonemia, the authors attempt to demonstrate that there must be additional factors that could be responsible for increasing glucagon secretion in the diabetic state. Thus, they look at paracrine signaling from delta and beta cells. Here the experimental strategy relies on determining changes in the responses to exogenously applied signaling molecules such as somatostatin and insulin.

As detailed below, this second part is less convincing. As a consequence, the results do not support the strong conclusion drawn in the manuscript that paracrine inhibition of alpha cells is disturbed in type 2 diabetes. An additional effort is required to be able to conclude that paracrine signaling is defective. Below is a list of suggestions and comments that could help improving this otherwise fine manuscript.

We agree that the second part was less convincing. We have added extensive new data to strengthen the conclusion that paracrine inhibition is impaired in T2D. Data from intact islets now show that pharmacological block of INSR or SSTR2 results in glucose dependence that is identical that in T2D islets, or in dispersed ND α -cells (Fig 4A-C). We further show in human pancreatic sections that SSTR2 is expressed mostly in the plasma membrane, whereas it is reduced and internalized in T2D (Fig 4D-F). Additional data show that insulin and somatostatin have little effect at low glucose, and we extend the membrane potential recordings to show insulin effects.

(1) The authors convincingly show that in type 2 diabetes glucose-dependent granule exocytosis is reduced in alpha cells. This is actually in line with a recent study showing that, contrary to widespread expectations, glucagon secretion is reduced in type 1 diabetes (Brissova et al., 2018, Cell Rep 22:2667). These are important results. It is still interesting (and also important) to study if paracrine interactions are affected in the islet in type 2 diabetes. However, as the authors probably will agree, paracrine signaling must be investigated differently and more thoroughly than in the manuscript in its current form. The experimental design used is not adequate because it only probes for receptor responses. It does not reveal endogenous paracrine activation (or lack thereof). What is required is to study these responses in intact islets and challenge intrinsic signaling with receptor antagonists or other pharmacological tools.

As requested, data from intact islets now show that pharmacological block of INSR or SSTR2 results in glucose dependence that is identical that in T2D islets, or in dispersed ND α -cells (Fig 4A-C. We further show in human pancreatic sections that SSTR2 is expressed mostly in the plasma membrane, whereas it is reduced and internalized in T2D (Fig 4D-F).

(2) If glucose-dependent glucagon secretion from intact islets were increased despite diminished exocytosis in isolated alpha cells, the authors could invoke other mechanisms such as loss of inhibitory paracrine signaling. I don't think that changes (e.g. increases) in glucagon secretion have been determined in islets from type 2 diabetic donors. This information is important for this paper.

Islet data are now included, see above.

(3) It would also be important to determine if receptor expression is downregulated. With many databases now available, this could be readily checked, and the data be included in the paper. By the way, a quick look at the Sandberg atlas does not show a reduction of somatostatin SSTR2 receptor expression in type 2 diabetes (<http://sandberg.cmb.ki.se/pancreas/>).

We now include immunostaining of human pancreatic sections, which clearly show that the majority of SSTR2 is internalized in T2D islets (both a- and b-cells), whereas it is primarily in the plasma membrane in ND islets.

[REDACTED]

(4) The studies addressing paracrine signaling also require determining how insulin and somatostatin secretion change in the same type 2 preparations. In this context, it would be important to have an overarching hypothesis about what is going on. What are the putative mechanisms through which paracrine signaling is diminished in type 2 diabetes?

We can only speculate that the same mechanisms that make peripheral tissue insensitive to insulin are at play also to make α -cells insensitive to insulin and somatostatin. Hypersecretion of insulin is a feature of developing T2D. Since β - and δ -cells are simultaneously activated, somatostatin may also be increased in this situation. The latter could then lead to SSTR2 desensitization in α -cells. Interestingly, the excessive internalization of SSTR2 we observe in T2D is consistent with tuning of SSTR2 surface expression by endosomal trafficking in the pituitary (Alshafie J Cell Biol 2020).

(5) More experimental detail is needed to be able to follow the figures. For instance, the authors should

state how the concentrations for the different stimuli were selected, for how long the cells were incubated in the different glucose concentrations, and what statistical tests were used. It is not clear from the Methods what protocols were used for the different studies. The use of the Student t-test is not adequate for multiple comparisons (see Figure 3d, e). In general, there is negligence with data analysis and presentation. This should be improved.

Thank you for pointing this out. We have improved the presentation and description of protocols with this comment in mind. Glucose incubation were at steady state (at least 20 min pre-incubation before imaging the first cell of a coverslip). We now state that t-tests were used to compare ND with T2D, and ANOVA for multiple testing (eg paracrine factors).

(6) In Figure 4e, f, the ordinate is wrongly labeled. It should be “action potential frequency”, not “membrane potential”.

Thank you. It should be “average membrane potential”, which is now corrected in the text.

(7) References 39 and 42 are the same.
corrected

Reviewers' Comments:

Reviewer #1:

Remarks to the Author:

These workers have addressed my comments to my satisfaction. I have no further asks. The additional whole islet data that they did alpha cell TIRFM is a nice addition.

reviewed by : Herb Gaisano

Reviewer #2:

Remarks to the Author:

The authors have added a substantial amount of convincing data that satisfactorily address the concerns I raised. The manuscript now includes a complete new set of experiments using intact human islets (new Figure 4). What is shown in this figure is really important and, together with the rest of the paper, will have an impact in the field of islet biology and diabetes.